# Early Pathogenesis of Wesselsbron Disease in Pregnant Ewes

**DOI:** 10.3390/pathogens9050373

**Published:** 2020-05-13

**Authors:** Judith Oymans, Lucien van Keulen, Paul J. Wichgers Schreur, Jeroen Kortekaas

**Affiliations:** 1Department of Virology, Wageningen Bioveterinary Research, Houtribweg 39, 8221 RA Lelystad, The Netherlands; judith.oymans@wur.nl (J.O.); lucien.vankeulen@wur.nl (L.v.K.); paul.wichgersschreur@wur.nl (P.J.W.S.); 2Laboratory of Virology, Wageningen University & Research, P.O. Box 16, 6700 AA Wageningen, The Netherlands

**Keywords:** Wesselsbron virus, flavivirus, vertical transmission, pregnant ewes, immunohistochemistry, neuroinvasion

## Abstract

Wesselsbron virus (WSLV) is a neglected, mosquito-borne flavivirus that is endemic to the African continent. The virus is teratogenic to ruminants and causes a self-limiting febrile illness in humans. Wesselsbron disease manifests with similar clinical signs and occurs in the same areas under the same climatic conditions as Rift Valley fever, which is therefore included in the differential diagnosis. Although the gross pathology of WSLV infection in pregnant ewes is reported in literature, the pathogenesis that leads to stillbirths, congenital malformations and abortion has remained undescribed. In the present study, pregnant ewes were inoculated with WSLV and subjected to detailed clinical- and histopathology 8 days later. The virus was mainly detected in foetal trophoblasts of the placenta and in neural progenitor cells, differentiated neurons, oligodendrocytes, microglia and astrocytes. Our study demonstrates that WSLV efficiently crosses the maternal–foetal interface and is highly neuroinvasive in the ovine foetus.

## 1. Introduction

In the past two decades, the world has experienced a remarkable increase in the emergence and re-emergence of arthropod-borne viruses (arboviruses), several of which belong to the family *Flaviviridae*, genus *Flavivirus*. Examples include West Nile virus (WNV), which emerged in New York city in 1999, and has become the leading cause of epidemic meningoencephalitis in the US. WNV is also (re)emerging and spreading northwards in Europe, being detected for the first time in Germany in 2018 [1,2,3]. In 2015, Zika virus (ZIKV) emerged in South America, resulting in an unprecedented outbreak, manifesting with congenital brain abnormalities [4]. Whereas the impact of WNV and ZIKV on human health is currently well recognized, it is important to remember that both viruses were largely neglected until two decades ago. The (re)emergence of arboviruses is stimulated by rapidly growing human and animal populations, intensified travel, trade and climate change. Particularly, increase in temperature and humidity may stimulate arthropod vectors to slowly move into new territories [5]. Due to their ability to affect both animals and humans, arboviruses with zoonotic potential require special attention. 

A neglected zoonotic arbovirus is Wesselsbron virus (WSLV). WSLV was first isolated from the brain and liver of a decomposed lamb in the Wesselsbron district in South Africa in 1955 [6]. The next month, the same virus was isolated from a human and mosquitoes of the *Aedes* genus, also in South Africa [7]. Since then, WSLV has been detected throughout the African continent either by virus isolation from vertebrates and mosquitoes or through detection of antibodies [6,8,9]. WLSV infects a wide range of domesticated animals like sheep, goats, cattle, camels and horses [8,10,11]. In 2013, WSLV was isolated from a black rat in Senegal, indicating that small rodents may also play a role in the maintenance of the virus [12]. Since the first isolation of WSLV from a human case in 1955, 33 human cases have been described, more than half of which were associated with laboratory exposure. These infections were associated with fever, headaches, myalgia and arthralgia [8,12,13]. Encephalitis as a result of WSLV infection was recorded once, when a person became infected after accidentally spraying a virus suspension into the eye [8]. Considering that there is little to no surveillance of WSLV in hospitals, prevalence of the infection in humans is almost certainly underestimated. 

Sheep seem to be the most susceptible to WSLV infection [8]. The infection in adult sheep remains asymptomatic or manifests with a mild-to-moderate fever [14,15]. In newborn lambs, the disease is more severe and can lead to death, within 3 days in 35% of cases, while older lamb are less susceptible [8,16]. In pregnant ewes, the infection may result in abortion or congenital malformations [17]. Developmental abnormalities include various malformations of the central nervous system (CNS), including hydranencephaly and muscular malformations (arthrogryposis). In goats and calves, congenital malformations and abortions seem to be less common [18]. 

Although gross pathology resulting from WSLV infection during ovine gestation was already reported in literature [8,17], the pathogenic events that result in vertical transmission and congenital malformations have remained undescribed. Insight into the pathogenesis of WSLV disease may facilitate the development of control tools, including vaccines, and may also improve our understanding of the pathology of related (zoonotic) neuroteratogenic flaviviruses. 

In the present study, ewes were inoculated with WSLV at one-third of gestation. With the aim to identify primary and secondary target cells and tissues, ewes were euthanized and necropsied 8 days after inoculation. Organs of the ewes and foetuses were evaluated by (histo)pathology, and the presence of WSLV was evaluated by reverse transcription quantitative PCR (RT-qPCR) and immunohistochemistry (IHC). Inoculation resulted in viremia in all inoculated ewes. Importantly, whereas no virus was detected in liver and spleen samples collected at necropsy, 8 days post inoculation, WSLV was shown to replicate efficiently in placental and foetal tissues. Immunohistochemistry illustrated that WSLV is highly neurotropic, neuroinvasive and neurovirulent in the ovine foetal CNS, targeting both neurons and neuroglial cells.

## 2. Results

### 2.1. Clinical Manifestation after Experimental WSLV Inoculation

To identify primary target cells of WSLV in pregnant ewes, ten ewes at 54 days of gestation were randomly divided over two groups. After a week of acclimatisation, at day 61 of gestation, one group was inoculated with WSLV (10^6.7^ TCID_50_) and the other group was mock-inoculated with medium. Rectal temperatures were measured and plasma samples were taken daily (Figure 1A). Surprisingly, no fever was measured in the WSLV-inoculated ewes (Figure 1B). However, viremia, as determined by detection of viral RNA, was observed during the first five days following infection (Figure 1C). At 8 days post inoculation, the ewes were euthanized and necropsies were performed. Samples were taken from the liver, spleen and the iliac and inguinal lymph nodes (LN), which drain the placenta. No macroscopic abnormalities were observed during necropsy, and all organ samples were negative for WSLV RNA (Figure 1D), suggesting WSLV is cleared rapidly from the blood and organs of the ewes.

### 2.2. WSLV Replicates in the Ovine Placenta 

The ovine placenta comprises around 30–60 placentomes connected by an intercotelydonary membrane. Each placentome contains maternal and foetal tissues, which are separated by several cell layers. The maternal epithelium and foetal trophoblasts form the actual barrier between mother and foetus in the synepitheliochorial part of the placenta, whereas in the so-called haemophagous zones, foetal trophoblasts are in direct contact with stagnant pools of maternal blood [19]. 

During necropsy, samples were taken from three placentomes from different parts of the placenta and from the intercotelydonary membrane of each foetus. Both the placentomes and the intercotelydonary membrane were shown to contain high levels of viral RNA (Figure 2A). Haematoxylin and eosin (HE) staining of the placentomes showed small areas with necrosis of the maternal epithelial cells (Figure 2B top). To detect WSLV by immunohistochemistry, a polyclonal antiserum was raised by immunizing rabbits with baculovirus-produced WSLV NS1 protein. Immunohistochemistry with this antiserum revealed positive staining of the necrotic areas (Figure 2B bottom). Foci of WSLV antigen were distributed across the whole placentome in all tested placentomes (Figure 2C) and consisted of infected foetal trophoblasts surrounding infected maternal epithelial cells (Figure 2D). WSLV antigen was also detected in foetal trophoblasts of the haemophagous zones (Figure 2E). The presence of high viral RNA loads in the placenta and absence of viral RNA in other organ samples collected at 8 days post inoculation demonstrates that the ovine placenta is a primary target organ of WSLV. The observed necrosis in the maternal epithelium suggests WSLV first replicates in maternal epithelial cells, after which foetal trophoblasts are targeted in the foetal part of the placenta.

### 2.3. WLSV is Highly Neurotropic and Neuroinvasive in the Ovine Foetus

Necropsy of the foetuses revealed no macroscopic differences between the foetuses of the WSLV-infected group and the control group. RT-qPCR analysis of foetal liver, brain, amniotic fluid, umbilical cord and plasma samples revealed the presence of high levels of viral RNA, except for the amniotic fluid samples (Figure 3A). WSLV antigen was not detected by IHC in the umbilical cord, suggesting that the viral RNA that was detected in umbilical cord homogenates originated from the foetal blood. WSLV was detected in foetal hepatocytes (Figure 3B) and sporadically in striated muscle fibres and osteoblasts (Figure 3C,D). 

Most interestingly, WSLV antigen was immunohistochemically detected throughout the foetal brain, including the telencephalon, diencephalon, mesencephalon, metencephalon and myelencephalon (Figure 4A). Infected differentiated neurons were observed in all parts of the brain, such as the pyramidal cells in the cortical plate, purkinje/granule cells in the cerebellar cortex, motoric neurons in the brain stem and pyramidal neurons of the hippocampus (Figure 4B–E). In addition, WSLV antigen was found in the ventricular and subventricular zone where radial glial cells and neural progenitor cells divide and proliferate (Figure 4F). Infection of nondifferentiated neuroblasts and glial precursor cells migrating from the (sub)ventricular zone to the cortical plate caused an abundant staining for WSLV in the intermediate zone (Figure 4G). 

To specifically identify the different cell types in the brain that are targeted by WLSV, double immunofluorescent stainings were set up in which various cell markers and WSLV antigen can be detected simultaneously. These experiments confirmed that foetal neurons are the primary target cells of WSLV (Figure 5). In addition, WSLV antigen was detected in all types of neuroglial cells: oligodendrocytes, microglia and astrocytes (Figure 5). Microglial cells in WSLV-infected foetuses had adopted a rounded, amoeboid shape in contrast to the ramified microglia found in mock-inoculated foetuses, suggesting activation of microglia following neuroinvasion by WSLV (Figure 6A,B). Activated microglia were seen invading sites of WSLV infection in the brain (Figure 6C,D), leading to the formation of microglial nodules throughout the brain.

### 2.4. No Evidence of WSLV Replication in Human Term Placental Explants

As WSLV is a zoonotic virus with the ability to transmit vertically in ruminants, we were interested in the ability of WSLV to infect human placental explants. To validate our protocol, ZIKV was taken along as a positive control, as it has been described in literature that ZIKV is able to infect term placental explants [20]. Term placental explants were incubated with WSLV, ZIKV or culture medium only, and samples were taken at 2, 4 and 6 days post inoculation. Although ZIKV was able to infect the placenta explants at all time points and showed slight growth over time, no evidence of WSLV infection was observed (Figure 7). 

As WSLV showed to be neurotropic, we further studied its zoonotic potential by investigating the possible capability of WSLV to infect human brain cells. Three human glioblastoma cell lines and an astrocyte cell line were incubated with WSLV, after which the monolayers were fixed and stained for WSLV antigen (Appendix A). The glioblastoma cell lines were highly permissive for WSLV, however, infection of astrocytes appeared to be limited.

## 3. Discussion

WSL disease was first described in 1956 by Weiss, Haig and Alexander [6]. These authors reported the association of WSLV infection with congenital malformations, stillbirths and abortions in sheep flocks and the isolation of the virus from the brain of a dead lamb. Although further observations from the field and from controlled animal experiments have confirmed the teratogenic potential of WSLV in ruminants, the pathogenesis of the disease has remained undescribed. The present study was performed to identify primary target cells and tissues with the aim to understand how WSLV crosses the ovine placental barrier and how the virus invades the foetal brain.

Inoculation of European-breed ewes at one-third of gestation did not result in febrile reactions or clinical signs within 8 days post inoculation, suggesting that WSLV infections in the field, at least in this breed of sheep, may remain unapparent until pregnancy is compromised. Moreover, at necropsy, no gross pathology in both the ewes and foetuses was observed. Despite the absence of macroscopic lesions or viral RNA in maternal liver and spleen samples at 8 days post inoculation, high levels of viral RNA were detected in placental samples. Immunohistochemical analysis revealed WSLV antigen in maternal epithelial cells, although staining was much more pronounced in foetal trophoblasts. The observation that areas of necrosis in the maternal epithelium were surrounded by infected foetal trophoblasts suggests that WSLV infects the maternal epithelium first, followed by infection of foetal trophoblasts. In addition to foetal trophoblasts lining the maternal epithelium, we found that WSLV infects trophoblasts in the haemophagous zones as well, representing a second route of vertical transmission. Remarkably, the presence of widespread, albeit focal, viral replication in the placenta was not associated with inflammation, and placentas appeared healthy. 

Immunohistochemical analysis of the foetal organs revealed WSLV antigen in the liver and, sporadically, in foetal striated muscle fibres and osteoblasts. Although no cytopathic effect was observed, viral replication in these cells may compromise muscle and bone development. The most striking finding of this study was the strong staining of WSLV antigen throughout the foetal brain, including the cerebral cortex, cerebellum and brain stem, with antigen detection in neurons, glial cells and neural progenitor cells. Whereas WSLV antigen in microglia may have resulted from infection, staining of these cells may also represent phagocytosed virus. The latter is supported by our finding that microglia in WSLV-affected foetal brains had adopted the activated, amoeboid phenotype and were found in large numbers in proximity to WSLV-affected areas. Notably, our finding that not only neural progenitor cells but also differentiated cells are targeted by WSLV suggests that the foetal brain is also susceptible at later time points of gestation. The apparent absence of lesions could suggest that WSLV replicates in the foetal brain without causing significant pathology. However, considering the previous reports of WSLV-associated embryonic, foetal or neonatal death and teratogenic defects, we consider it unlikely that the foetuses of the present work would have developed normally, had pregnancy been allowed to continue. 

Despite the severe consequences of infection during pregnancy and the widespread distribution of WSLV in Africa, the virus has not been associated with significant outbreaks in endemic areas. This is possibly explained by herd immunity resulting from exposure to the virus outside gestation and protection of newborns via maternally-derived antibodies. An introduction of WSLV into an immunologically naive population, particularly during the breeding season, could however result in rapid dissemination and unprecedented disease manifestation. Such “virgin soil” arbovirus outbreaks tend to be severe, as previously exemplified by the Bluetongue and Schmallenberg outbreaks in Europe and the West Nile virus (WNV) and ZIKV outbreaks in the Americas [21,22,23,24]. It is therefore important to further assess WSLV virulence for animals and humans and its ability to be disseminated by mosquito vectors indigenous to currently unaffected areas. 

## 4. Materials and Methods 

### 4.1. Viruses and Cells

WSLV, strain SAH177 passage 14 on Vero cells and originally isolated from a human in 1955 in South Africa [7], was kindly provided by prof. Janusz Paweska of the National Institute for Communicable Diseases of the National Health Laboratory Service (NICD-NHLS), Sandringham, South Africa. ZIKV strain PRVABC59, originally isolated from human serum in Puerto Rico in 2015, was obtained from American Type Culture Collection (ATCC; VR-1843). Virus stocks were prepared by inoculating Vero-E6 cells at low multiplicity of infection (MOI; 0.01). Vero-E6 cells were obtained from ATCC (CRL-1586) and maintained in minimal essential medium (MEM; Gibco, Thermo Fischer Scientific, Breda, The Netherlands) supplemented with 5% foetal bovine serum (FBS; Gibco), 1% antibiotic/antimycotic (a/a; Gibco), 1% nonessential amino acids (Gibco) and 1% L-glutamine (Gibco) (complete medium) at 37 °C with 5% CO_2_. Virus stocks were titrated by incubating Vero E6 cells with serial dilutions of the virus for 6 days, after which cytopathic effect (CPE) was observed and titres were calculated using the Spearman–Kärber algorithm [25,26]. Both WSLV and ZIKV were passaged twice on Vero-E6 cells at our institute, after which they were used in experiments described below.

Sf9ET cells (ATCC^®^ CRL-3357™) were cultured in Insect-XPRESS medium (Lonza, Maastricht, the Netherlands) supplemented with 1% a/a. High Five cells were maintained in Express Five medium (Gibco) supplemented with 1% glutamine and 1% a/a. Both insects’ cell lines were cultured in suspension at 28 °C in a shaking incubator. 

### 4.2. Experiments with Explants from Human Term Placentas

Term placentas (n = 2), obtained after caesarean section, were received from the Isala hospital (Zwolle, the Netherlands). Placentas were transported on ice, after which explants of 4 × 4 mm were cut from the chorionic villi of the placenta. Each explant was washed three times in PBS supplemented with 1% a/a and placed in a well in a 24-well plate with 1 mL complete medium (40% Dulbecco’s modified Eagle medium (DMEM; Gibco), 40% F12 nutrient mixture (Gibco), 10% FBS and 1% a/a). Explants (n = 4) were subsequently inoculated with 2.5 × 10^5^ TCID_50_/mL ZIKV or WSLV. At 16 hours post infection, medium was refreshed, and at 2, 4 and 6 days post infection, explants were frozen at −80 °C until further processing.

### 4.3. Ethics Statement

The pregnant ewe trial was conducted in accordance with the Dutch Law on Animal Experiments (Wet op de Dierproeven, ID number BWBR0003081) and the European regulations on the protection of animals used for scientific purposes (EU directive 2010/63/EU). The procedures were approved by the animal ethics committee of Wageningen Bioveterinary Research (WBVR) and the Dutch Central Authority for Scientific Procedures on Animals (permit number AVD401002017894).

Human placentas were obtained after caesarean section of healthy women. Placentas are regarded as medical waste and therefore do not fall under the scientific medical research law of the Netherlands, and the experiments described in this manuscript do not require approval from an institutional review board. All donors have given written consent, and consent forms are stored in accordance with the Dutch privacy law.

### 4.4. Pregnant Ewe Trial

The pregnancies of 10 Texel–Swifter mix-breed ewes were synchronised by progesterone sponge treatment and natural mating at a conventional Dutch sheep farm. To confirm pregnancy, ultrasounds were performed at 6–7 weeks after mating. At day 54 of gestation, the ewes were transported to WBVR to acclimatise for 7 days until the start of the trial. 

At 61 days of gestation, the ewes were inoculated intravenously with 10^6.7^ TCID_50_ WSLV in 1 mL medium or with 1 mL medium (negative control animals). During the whole study, animals were monitored for clinical signs twice a day, and rectal body temperatures were measured once per day. EDTA blood samples were collected daily from the day of inoculation. At 8 days post inoculation, the ewes were euthanised by intravenous administration of 50 mg/kg sodium pentobarbital (Euthasol®, ASTfarma, Oudewater, Netherlands) and subsequent exsanguination. Foetuses were exsanguinated as well by cutting the umbilical cord, and foetal blood was collected in EDTA tubes. At the necropsy of each ewe, samples were taken from the liver, spleen, iliac lymph node and inguinal lymph node. From the foetuses, samples were taken from the liver, brain and leg muscle/bone marrow of the femur. Samples from three placentomes and the intercotelydonary membrane were taken per placenta, and samples were taken from the umbilical cord and amniotic fluid. All samples were collected in duplicate, one sample was placed on dry ice and stored at −80 °C, while the other sample was fixed in 10% neutral buffered formalin for 48 h for histology and immunohistochemistry. Formalin-fixed samples were processed routinely into paraffin blocks.

### 4.5. Detection of Viral RNA

Organ samples from the pregnant ewe trial were homogenised by mixing 0.3–1 g of tissue with 7 mL complete medium in IKA Ultra Turrax DT-20 tubes. Cell debris was removed by centrifugation for 15 min at 4952 x g in 15 mL Falcon tubes. Organ suspension or plasma (500 µL) was added to 2.5 mL NucliSENS easyMAG Lysis Buffer (Biomérieux, Marcy-l’Étoile, France), after which RNA was extracted using the NucliSENS easyMAG (Biomérieux) according to manufacturer’s protocol.

To extract RNA from the human placental explants, the explants were lysed in Lysing Matric D tubes (MP Biomedicals) in 1 mL TRIzol Reagent (Invitrogen). Homogenised suspensions were made using the TeSeE Precess 24 bead beater for 2 × 23 s at 6500 RPM. RNA was isolated from 350 µL suspension using the Direct-zol RNA miniprep kit (Zymo Research, Irvine, CA, USA) according to manufacturer’s protocol.

Primers and probes were designed using the PrimerQuest Tool for Integrated DNA technologies (IDT, Leuven, Belgium) that target the NS5 protein of WSLV. Primers and probes with the following sequences were purchased from IDT: forward primer: 5’-GGA CCA TGA AAG TGT TGG-3’, reverse primer: 5’-CAA TCA CAT CTG GAT AGG-3’; Probe: 5’-6FAM- TGA ACG ATG GAA ACA CGT GAA CAC AGA TMR-3’. ZIKV primers and probes were used as described by Lanciotti et al. [27]. Five µL of the RNA was used in a RT-qPCR using the LightCycler RNA Amplification Kit HybProbe (Roche, Almere, the Netherlands). Cycling conditions were as follows: 45 °C for 30 min, 95 °C for 5 min, 45 cycles of 5 s at 95 °C and 35 s at 57 °C, followed by cooling down to 30 °C. 

WSLV and ZIKV standards for quantification were made by isolating RNA of a 100× diluted virus stock. Virus was lysed in Trizol, and RNA was isolated using the Direct-zol RNA miniprep kit (Zymo Research) according to the manufacturer’s protocol. A 10x dilution series was made of the isolated RNA in H_2_O, and the dilution series was aliquoted for PCR runs. As the TCID_50_ titre of the virus stock was known, this standard was used to quantify the RNA as TCID_50_ equivalent/mL. 

### 4.6. Production of Polyclonal Rabbit Sera against WSLV NS1

The sequence of the NS1 protein of WSLV, NCBI GenBank number: JN226796.1, was used to develop a DNA construct, encoding this gene, flanked by a N-terminal GP64 signal sequence and a C-terminal twin Strep-Tag. The resulting construct was synthesised by GenScript (Piscataway, NJ, USA). The gene with flanking sequences was subsequently cloned into a pBAC-3 baculovirus vector, after which a recombinant baculovirus was produced using the flashBAC ULTRA baculovirus expression system (Oxford Expression Technologies, Oxford, UK). Transfection mixtures containing the pBAC-3 vector, a bacmid and celfectin II were added to wells of a 6-well plate, each containing 1,000,000 SF-9 ET cells. After successful rescue of the baculovirus, stocks were prepared by inoculating SF-9 ET suspension cultures at low MOI (<0.1). The NS1 protein was produced by infecting High Five cells at high MOI (>5) according to manufacturer’s protocol (Thermoscientific, Rockferd, IL, USA). Strep-Tactin resin (IBA) columns were used to purify the proteins from the supernatants, after which buffers were exchanged to Tris-buffered saline in Amicon Ultra centrifugal filters (Merck). Purity of the NS1 protein was assessed in a 4–12% SDS gel stained with GelCode Blue stain reagent (Thermoscientific). 

Polyclonal rabbit sera against WSLV NS1 were produced by GenScript (Piscataway, NJ, USA) by immunizing two New Zealand rabbits with 1 mg of purified NS1 following their standard protocol for polyclonal antibody production.

### 4.7. Histology and Immunohistochemistry

Paraffin-embedded tissues were cut into 4 μm sections, collected on silane-coated glass slides and dried for at least 48 h at 37 °C. After deparaffinisation and rehydration in graded alcohols, sections were either stained routinely with haematoxylin and eosin (HE) or immunostained. For immunostaining, endogenous peroxidase was blocked for 30 min in methanol/H_2_O_2,_ followed by antigen retrieval by autoclaving the slides in pH 6 citrate buffer (Antigen unmasking solution, Vector Laboratories) for 15 min. The WSLV NS1 polyclonal antiserum that was produced as described above was used at a dilution of 1:1000 to detect WSLV NS1. Antibodies to cell markers were used to visualize neurons (anti-Neurofilament 200, Sigma-Aldrich, St. Louis, MO, USA), astrocytes (anti Glial Fibrillary Acidic Protein, Sigma-Aldrich), oligodendrocytes (anti CNPase, Sigma Aldrich) and microglia (anti IBA-1, ITK Diagnostics, Uithoorn, Netherlands). Horseradish peroxidase (HRP; Invitrogen, Carlsbad, CA, USA) or Alkaline phosphatase (AP; Vector laboratories, Peterborough, UK) conjugated anti-mouse or anti-rabbit polymers were used as secondary antibodies, followed by incubation with NovaRed or VectorRed substrate respectively (Vector Laboratories). Sections were briefly counterstained with haematoxylin, dehydrated and mounted permanently. For immunofluorescent double staining, the HRP substrate was replaced by Alexa Fluor 488 or 546 tyramide reagent (Invitrogen) and sections were mounted in antifading mounting medium containing DAPI (Vector laboratories). Sections were photographed with an Olympus BX51 (fluorescence) microscope equipped with Cell D/Cell Sense software and a high-resolution digital camera. Overview images were constructed with the multiple image alignment tool of the Cell D software, and immunostaining was coloured red using the phase colour coding tool. Monochromatic digital photographs for immunofluorescence were false coloured in green for the Alex Fluor 488 dye and in red for the Alexa Fluor 546 dye. 

## Figures and Tables

**Figure 1 pathogens-09-00373-f001:**
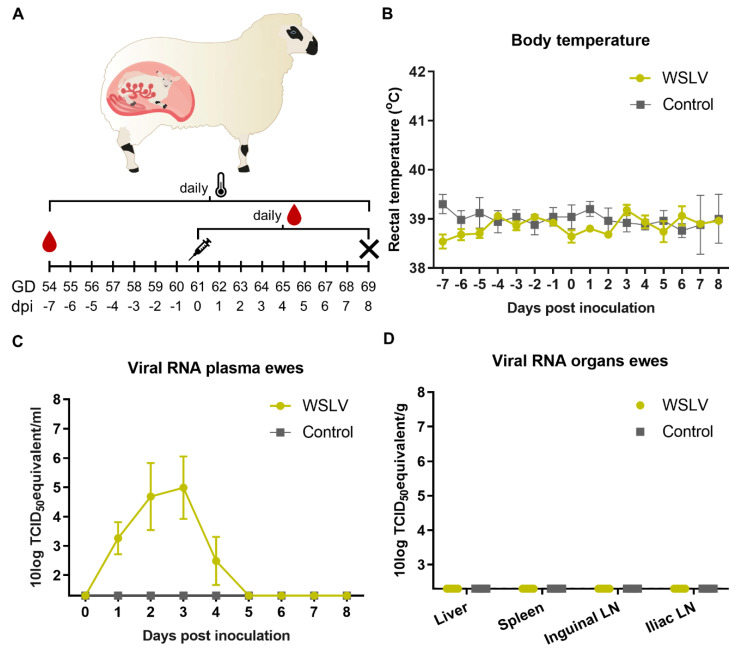
Wesselsbron virus (WSLV) infection in pregnant ewes. (**A**) Experimental set-up of pregnant ewe trial. Ewes were inoculated at gestation day (GD) 61. At 8 days post inoculation (dpi), the ewes were euthanized and necropsied. Rectal temperatures (**B**), viremia (**C**) and viral RNA in the organs of the ewe (**D**) are depicted. Error bars represent averages with standard deviation (SD).

**Figure 2 pathogens-09-00373-f002:**
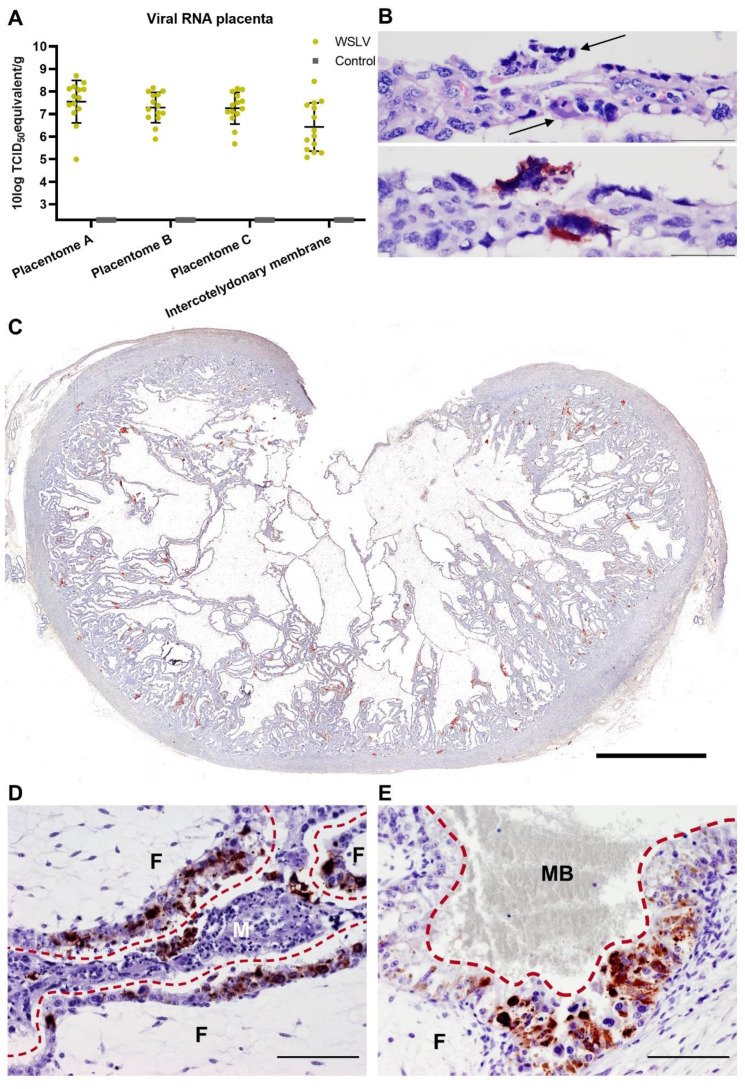
WSLV infection in the ovine placenta. (**A**) Detection of viral RNA in the placentas of WSLV-inoculated and control ewes. Samples were collected from three placentomes and the intercotelydonary membrane from each foetus. Bars represent averages with SD. (**B**) Top: HE (haematoxylin and eosin) staining of a maternal villus with necrotic syncytial epithelial cells (arrows). Bottom: Serial section showing the same cells staining positively for WSLV. (**C**) Overview of a WSLV infected placentome showing the distribution of WSLV-positive foci across the whole placentome. (**D**) Synepitheliochorial placenta showing positive staining for WSLV of the maternal epithelium and adjacent foetal trophoblasts. (**E**) Haemophagous zone of the placenta also showing strong staining of the foetal trophoblasts which are in direct contact with maternal blood (MB). The red interrupted lines indicate the barrier between the maternal (M) and foetal (F) part of the placenta. Bars are 50 μm (**B**), 5000 µm (**C**) or 100 µm (**D**,**E**).

**Figure 3 pathogens-09-00373-f003:**
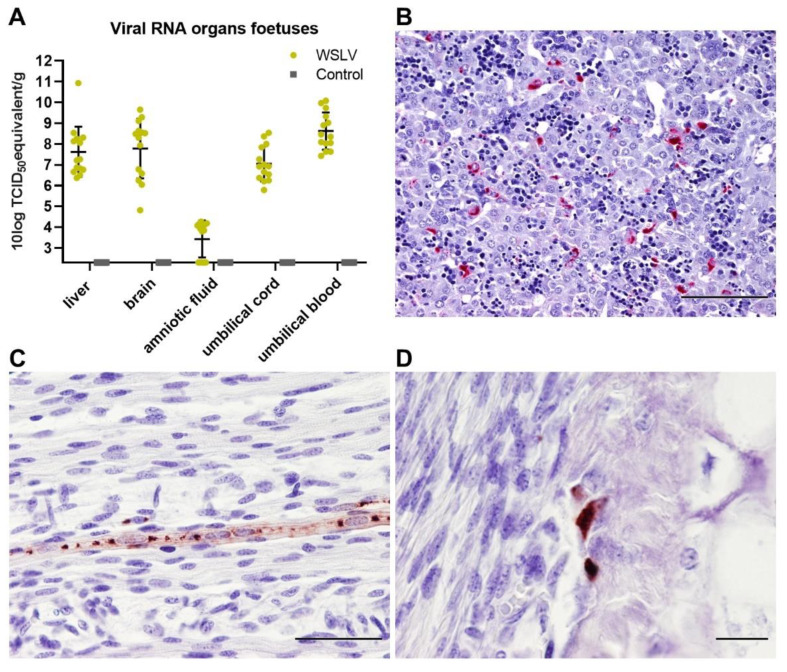
WSLV infection in foetal organs. (**A**) Viral RNA detected by PCR in the liver, brain, amniotic fluid, umbilical cord and plasma of the foetuses. Bars represent averages with SD. (**B**) Immunohistochemical staining for WSLV antigen in the foetal liver showing scattered infected hepatocytes. Notice the extramedullary haematopoiesis in the hepatic sinusoids, which is normal in the foetal liver at this stage. (**C**) WSLV-positive myocyte in the striated muscle of the hind leg. (**D**) WSLV-positive osteoblasts in the cortical bone tissue of the femur. Bars are 100 µm (**B**), 50 μm (**C**) or 20 μm (**D**).

**Figure 4 pathogens-09-00373-f004:**
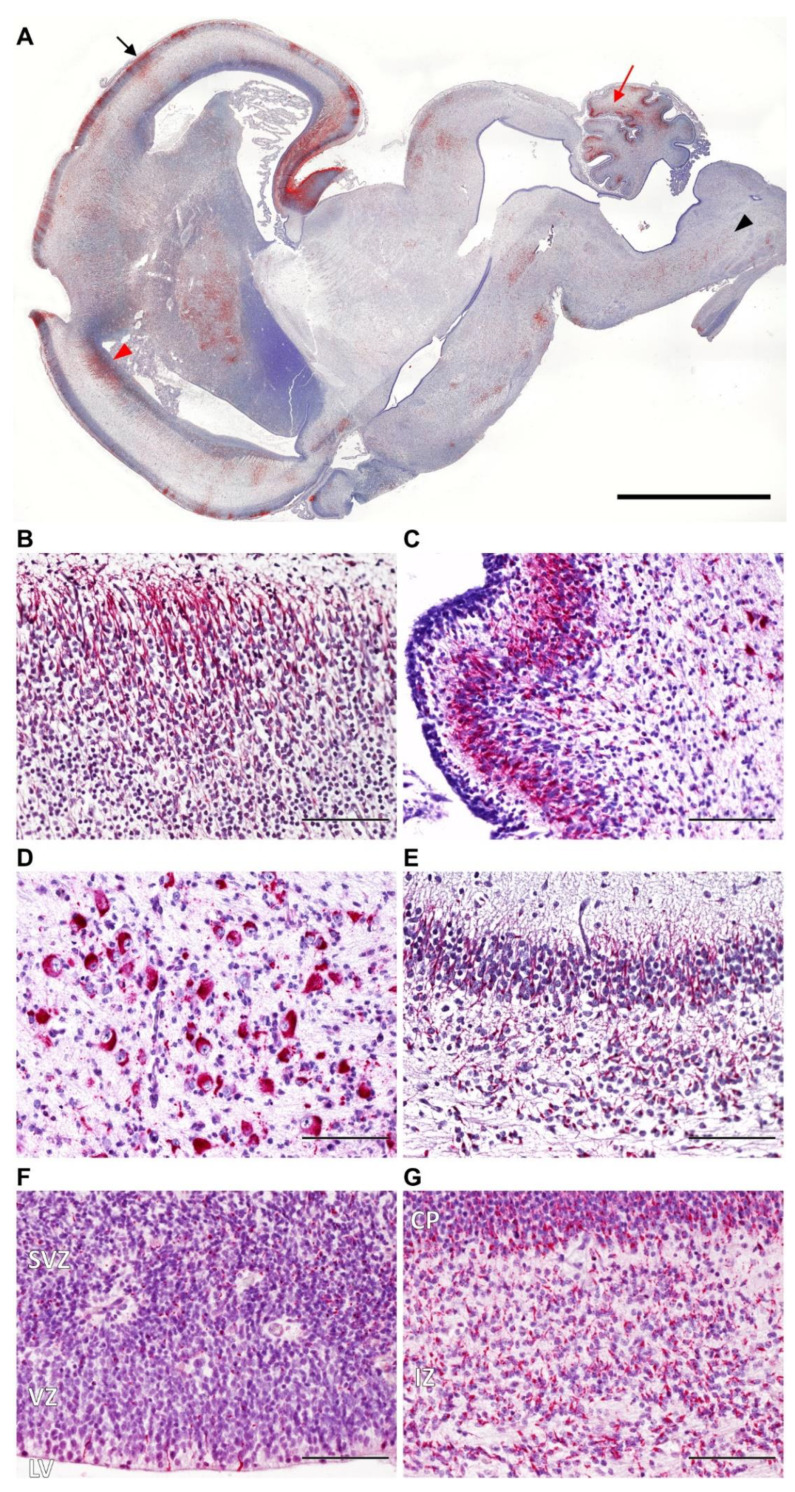
WSLV infection in the foetal brain. (**A**) Immunohistochemical detection of WSLV antigen in a longitudinal section of the foetal brain. WSLV antigen is present in all areas of the brain, including the cerebrum (black arrow), cerebellum (red arrow), brain stem (black arrowhead) and (sub)ventricular zone (red arrowhead). Detailed staining of WSLV antigen in the neurons of the cerebral cortical plate (**B**), cerebellar cortex (**C**), brain stem (**D**), hippocampus (**E**), (sub)ventricular zone (**F**) and intermediate zone (**G**). SVZ: subventricular zone, VZ: ventricular zone, LV: lateral ventricle, CP: cortical plate and IZ: intermediate zone. Bars are 5000 µm (**A**) or 100 µm (**B**–**G**).

**Figure 5 pathogens-09-00373-f005:**
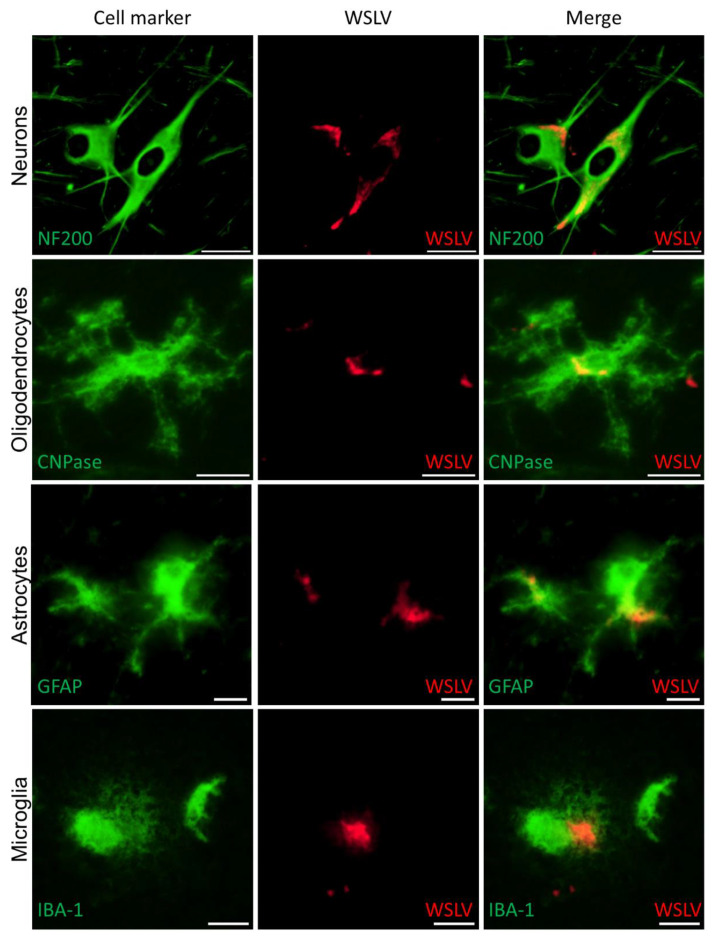
Identification of WSLV-infected foetal brain cells. Double immunofluorescent microscopy images of foetal brains to identify WSLV-infected cells. Cell markers are indicated in green, WSLV antigen in red. Neurons were visualised using anti-Neurofilament 200 (NF200), oligodendrocytes by anti CNPase, astrocytes using anti Glial Fibrillary Acidic Protein (GFAP) and microglia using anti IBA-1. Bars are 50 µm (neurons), 20µm (oligodendrocytes) or 10 µm (microglia and astrocytes).

**Figure 6 pathogens-09-00373-f006:**
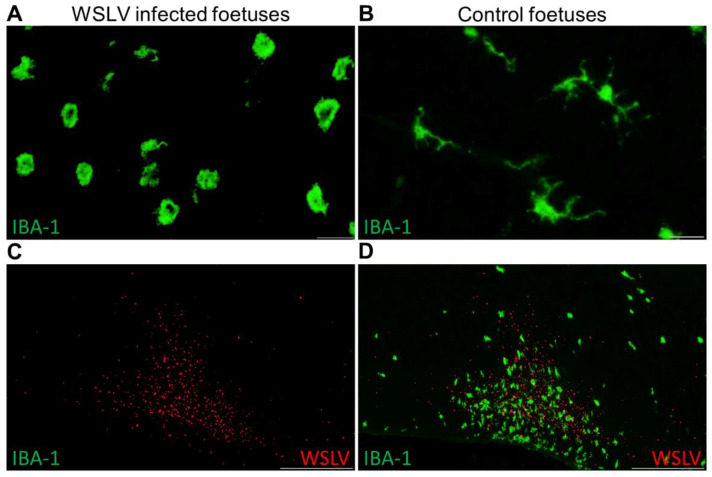
Activated microglia after WSLV infection. Specific staining of microglia (anti IBA-1) in the brain of WSLV-infected foetuses (**A**) and control foetuses (**B**). Note the morphological change of the microglia in the WSLV-infected brain from a resting ramified appearance to a more rounded shape. Foci of WSLV replication in the brain (**C**) were actively invaded by amoeboid microglia, leading to the formation of microglial nodules (**D**). Bars are 20 μm (**A**,**B**) or 500 μm (**C**,**D**).

**Figure 7 pathogens-09-00373-f007:**
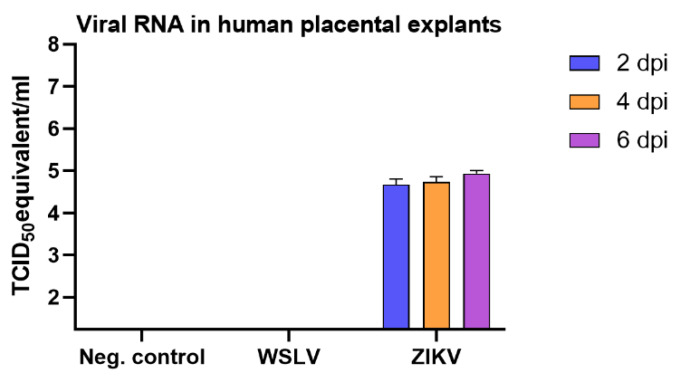
Susceptibility of human term placental explants for WSLV. Human term placentas were incubated with Zika virus (ZIKV), WSLV or culture medium only. At 2, 4 and 6 days post inoculation (dpi), samples were collected for RNA extraction (n = 4), after which viral RNA was measured by RT-qPCR.

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
