# Peer review of "Early Pathogenesis of Wesselsbron Disease in Pregnant Ewes"

_pathogens, 2020, doi:10.3390/pathogens9050373_

Round 1

Reviewer 1 Report

In this manuscript, Oymans et al., described the vertical transmission of Wesselsbron virus (WSLV) in the pregnant ewes at 8 day post-infection. The authors showed that WSLV was detected in foetal trophoblasts of the placenta and in neural progenitor cells, differentiated neurons, oligodendrocytes, microglia and astrocytes which were observed only by immunohistochemistry. Finally, the authors demonstrated that WSLV efficiently crossed the maternal-foetal interface and is highly neuroinvasive in the ovine foetus. Although this is a very interested and unique (but not novel) studies in terms of WSLV, the authors concluded the WSLV’s vertical transmission and neuroinvasive in the pregnant ewes only observed the clinical- and histopathological results at one time point (8 days post-infection). This conclusion is very preliminary without other histopathological evidence in the WSLV-infected pregnant ewes.

  1. The first section of Introduction is too long with no significant related to this study.
  2. Why did the authors choose evaluated the detailed clinical- and histopathology at 8 days post-infection?
  3. As for Figure 3: WSLV infection in foetal organs. I think that the authors should repeat this experiment and evaluate all the clinical and immunohistochemistry evidences at 4 or 6 days post-infection. Hence, it will convince us that WSLV indeed highly crosses maternal-foetal interface during the infection cycle in the pregnant ewes.
  4. What is “WESSV” in the Figure 7?

Reviewer 2 Report

The manuscript by Oymans et al. describes the infection of pregnant ewes with Wesselbron virus. The article targets the interesting question of the intrauterine infection with this virus and illustrates that the infection via the placenta starts with maternal epithelial cells and then infected the foetal tissues with special emphasis on neuronal tissues.
The experiments described are sound and the conclusions draw well justified. The whole study is of a purely descriptive nature. However, it would have been interesting to be able to draw more functional conclusion from the experiments as for example the question of the functionality of the infected neurons or if microglia can contain the virus infection after phagocytosis. What would be the role of Astrocytes during infection? Are they activated to expresses Interferon? Could interferon induction explain the reduced infection of human astrocytes? Nevertheless this manuscript offers important new finding that can be used to further investigate the interaction of Wesselbron virus with its ovine host.
Minor comments:
Line 154 Figure legend 4. Why are the authors only talking about WSLV antigen in neurons in the different brain regions? In Figure 5 they clearly show that also other cell types are affected

Reviewer 3 Report

The authors have studied the infection of Wesselsbron virus in pregrnant ewes and showed that the virus is detected in foetal brain tissue. In this work it is clearly shown that virus goes through the placenta and infects directly the embryo. This may cause abortion though cellular damages at 8 dpi are not very extended.

There are very minor points.

In fig 2A the error bars for brain sample does'nt seem correct

In line 83 "WSLV is very neurotropic"

Round 2

Reviewer 1 Report

The authors have addressed my comments and revised the introduction sentences.

I think that it can be published in this present version.